# Application of CoLD-CoP to Detecting Competitively and Cooperatively Binding Ligands

**DOI:** 10.3390/biom14091136

**Published:** 2024-09-09

**Authors:** Shiva V. Patnala, Roberto Robles, David A. Snyder

**Affiliations:** Department of Chemistry, College of Science and Health, William Paterson University, 300 Pompton Road, Wayne, NJ 07470, USA; patnalasv@gmail.com (S.V.P.); roblesr1@student.wpunj.edu (R.R.)

**Keywords:** nuclear magnetic resonance (NMR), cooperative/competitive ligand binding, diffusion spectroscopy (DOSY), fragment-based drug discovery (FBDD)

## Abstract

NMR utilization in fragment-based drug discovery requires techniques to detect weakly binding fragments and to subsequently identify cooperatively binding fragments. Such cooperatively binding fragments can then be optimized or linked in order to develop viable drug candidates. Similarly, ligands or substrates that bind macromolecules (including enzymes) in competition with the endogenous ligand or substrate are valuable probes of macromolecular chemistry and function. The lengthy and costly process of identifying competitive or cooperative binding can be streamlined by coupling computational biochemistry and spectroscopy tools. The Clustering of Ligand Diffusion Coefficient Pairs (CoLD-CoP) method, previously developed by Snyder and co-workers, detects weakly binding ligands by analyzing pairs of diffusion spectra, obtained in the absence and the presence of a protein. We extended the CoLD-CoP method to analyze spectra pairs (each in the presence of a protein) with or without a critical ligand, to detect both competitive and cooperative binding.

## 1. Introduction

For at least two decades, a “target-rich”, “lead-poor” pipeline has characterized modern drug development [1]. Over the last 20 years, the amount by which pharmaceutical companies reallocate their net revenues into R&D has increased by about 20%, and by 2021, pharmaceutical companies collectively spent a third of their total budget on preclinical phase R&D [2]. Fragment-Based Drug Discovery (FBDD) addresses the lead generation problem by combinatorial assembly of (perhaps weakly) binding fragments [3]: the FBDD process identifies multiple, small “fragment” molecules, which are then assembled, like pieces of a jigsaw puzzle, and optimized to increase target hit affinity up to 100–1000-fold [4,5].

A key step in FBDD is, therefore, figuring out which (weakly) binding fragments are linkable, i.e., bind *cooperatively* so that linking those fragments would likely significantly enhance binding affinity [5]. On the other hand, a compound (full size or fragment) that binds *competitively* with the endogenous substrate or ligand of a given enzyme or receptor is a good lead for developing inhibitors of that enzyme or receptor [6]. Furthermore, competitive inhibitors are useful probes for better understanding enzyme active sites, chemistry, and even enzyme flexibility and dynamics [7].

Nuclear Magnetic Resonance (NMR) is a powerful tool with many applications in drug development, including FBDD pipelines [4,8]. NMR-based techniques, such as Diffusion Ordered Spectroscopy (DOSY), which identifies ligands by their change in diffusion coefficient upon target binding, can identify ligands with K_D_ values ranging from 100 pM to 10 mM [9,10,11]. However, promising fragment molecules may very well bind with a K_D_ up to 30 mM [12], and experiments to detect fragment binding typically require protein concentrations of one-thousandth or, depending on the type of experiment performed, up to one-tenth of the ligand concentration [13]. Challenges in the use of DOSY to identify ligand binding include the large uncertainty in diffusion coefficients inferred from DOSY data [14,15] as well as the inherent low sensitivity of NMR experiments. Ensuring enough protein is available to form protein–ligand complexes at concentrations sufficient to identify even the weakest-binding promising fragments typically requires around 10 mM ligand concentration and hence 10 μM protein concentration.

This paper demonstrates the application of the Clustering of Ligand Diffusion Coefficient Pairs (CoLD-CoP) method [16], which uses statistical techniques to overcome the large uncertainty in diffusion coefficients inferred from DOSY data, to identify competitively and cooperatively binding ligands. We demonstrate competitive binding on a well-studied system: 4-hydroxy-α-cinnamic acid (4-HCCA) and salicylic acid are each known to competitively inhibit (mushroom) tyrosinase [17,18] and hence are expected to compete with each other in binding that enzyme. On the other hand, lysozyme, a well-studied enzyme catalyzing the hydrolysis of a polymeric substance, has a large active site that can accommodate a macromolecular substrate. Hence, it would not be surprising to discover competitive inhibitors of that enzyme which cooperatively bind it. Imidazole [19,20], tris [21,22], and N-acetyl-glucosamine (GlcNAc) [23] are known competitive inhibitors of lysozyme. In particular, the substrate of lysozyme, peptidoglycan, is a co-polymer of GlcNAc and N-acetylmuramic acid [24]. We demonstrate that we can detect both competitive and cooperative binding—the former at a protein concentration of <5 μM and the latter being an expected result validated with Molecular Mechanics Generalized Born Surface Area (MM-GBSA) [25] calculations—using CoLD-CoP.

## 2. Materials and Methods

Figure 1 outlines the steps of our approach. Briefly, our approach involves making three different mixtures: one containing only the putative ligands (the ligand-only mixture: Step 1.A), one with putative ligands and a protein of interest (the ligand+protein mixture: Step I.B), and one with the putative ligands, a protein of interest, and a known ligand binding the protein of interest (the ligand+protein+known ligand mixture: Step I.C). Following the acquisition and processing of DOSY data for the ligand-only, ligand+protein, and ligand+protein+known ligand mixtures (Step 2), the DOSY peak lists are compared using CoLD-CoP [16].

As described in Figure 1, substituting the ligand+protein+known ligand peak list for the ligand-only peak list typically used in running CoLD-CoP (c.f. Step 3.B.i) identifies ligands diffusing more slowly in the absence of the known ligand (Step 3.B.ii), whereas substituting the ligand+protein peak list for the ligand-only peak list and the ligand+protein+known ligand peak list for the ligand+protein peak list identifies ligands diffusing more slowly in the presence of the known ligand (Step 3.B.iii). As illustrated in Figure 2, ligands that diffuse more slowly in the absence of the known ligand bind the protein of interest more effectively without the known ligand and hence compete with it for protein binding. Conversely, ligands that diffuse more slowly in the presence of a known ligand bind the protein of interest better with the known ligand, indicating cooperative binding.

Table 1 describes the components of the ligand-only, ligand+protein, and ligand+protein+known ligand mixtures used to validate the approach described in Figure 1. Briefly, solution TL is the ligand mixture used for probing competitive/cooperative binding of mushroom tyrosinase, with TLP being the corresponding ligand+protein mixture and TLPK being the corresponding ligand+protein+known ligand mixture. TLP was prepared by adding mushroom tyrosinase (to a concentration of 4 μM) to a portion of mixture TL; in the process of preparing mixture TLP, some salicylic acid precipitated, resulting in a concentration of only 1 mM in mixtures TLP and TLPK as opposed to the 4 mM concentration in mixture TL. TLPK was prepared by mixing a portion of mixture TLP with 4-HCCA.

Similarly, the ligand-only, ligand+protein, and ligand+protein+known ligand mixtures for probing competitive/cooperative lysozyme binding are labeled LL, LLP, and LLPK. LLP was prepared by adding lysozyme (to a concentration of 1 mM) to a portion of mixture LL, and LLPK was prepared by adding GlcNAc to mixture LLP. The solvent for all solutions was D_2_O. Solutions LL, LLP, and LLPK were adjusted to pH = 6 using NaOH. The pH of solutions TL, TLP, and TLPK was not adjusted.

Data acquisition was performed on a 400 MHz Bruker Avance^TM^ III using the ledbpgp2s pulse sequence with a diffusion time of 90 ms and a gradient pulse length of 1.9 ms. Initial processing (in the direct dimension) utilized NMRPipe [26], and DOSY processing proceeded using the DOSY Toolbox [27] along with the Covariance Toolbox [28], resulting in the DOSY spectra shown in Figure 1. Both DOSY processing and CoLD-CoP analysis were performed in MATLAB (various versions used including R2015b, R2022a, R2022b, R2023a, R2023b) [29]. iGEMDOCK [30] docked tris, imidazole, and GlcNAc to lysozyme (PDB ID 2D4K15). The ternary complex (tris-GlcNAc-lysozyme) was manually adjusted using the Schrödinger [31] interface. Complexes were energy minimized in DESMOND [32] using the Schrödinger interface and default minimization parameters. LigPlot+ [33,34] analyzed and illustrated ligand/protein interfaces. Structures were rendered in MacPYMOL [35]. MM-GBSA and MM-PBSA calculations were performed using the FastDRH webserver [36].

## 3. Results

### 3.1. Detecting Competitive Binding to Tyrosinase

Using the approach described in Figure 1, CoLD-CoP identifies two clusters of diffusion coefficient pairs as diffusing significantly slower in the absence of 4-HCCA than in its presence, indicating competitive binding (Figure 2). Based on the resonance assignments for mixture TL (Appendix A; Appendix A provide resonance assignments for mixtures TLP and TLPK, respectively), both of these clusters are assigned to salicylic acid. Thus, CoLD-CoP correctly identifies that 4-HCCA and salicylic acid competitively bind tyrosinase. Note, however, that due to their similar molecular masses and hence diffusion coefficients, the clustering routines in CoLD-CoP cannot distinguish between tris and tartaric acid; salicylate (which also has a similar molecular mass and hence diffusion coefficient) is distinguished (albeit split into two clusters) due to its competitive binding changing its diffusion coefficient in the presence of 4-HCCA.

### 3.2. Detecting Cooperative Binding to Lysozyme

CoLD-CoP identifies that two components of the ligand mixture LL with diffusion coefficients and chemical shifts corresponding to tris and imidazole diffuse slower in the presence of GlcNAc than in its absence (Figure 3B,C), indicating that tris and imidazole each bind cooperatively with GlcNAc. As shown in Figure 3A, even though tris and imidazole are each known competitive inhibitors of lysozyme, they do not compete with GlcNAc for binding lysozyme.

### 3.3. Validation of Cooperative Binding

The cooperative binding of tris and GlcNAc, though unexpected, is not surprising given that lysozyme’s active site is large enough to accommodate a polymeric substrate. Modeling the ternary complex (coordinates in PDB format available as Appendix A) of lysozyme, tris, and GlcNAc (Figure 4 and Figure 5) illustrates that in this complex, tris and GlcNAc hydrogen bond, providing a potential mechanism for the cooperative binding observed in our CoLD-CoP-based assay. MM-GBSA calculations further validate cooperative binding; note that, at least for parameter sets correctly identifying GlcNAc as a better binder than tris, the binding energy for GlcNAc and tris together is more negative than the sum of the binding energies for each (Table 2).

The dimerization of lysozyme [20,37] explains the cooperative binding of imidazole (which can bind lysozyme allosterically [20]) and GlcNAc: each bind to monomeric lysozyme, and GlcNAc binding facilitates imidazole binding by shifting lysozyme’s dimerization equilibrium toward the monomer (Figure 3).

## 4. Discussion

A putative ligand identified as binding competitively with a lead fragment (in this example, 4-HCCA and GlcNAc) would be a poor choice to assemble with the lead fragment in the context of FBDD. However, the identification of competitive binders, which may themselves be agonists, to endogenous ligands/substrates provides protein chemists with probes of protein structure and function [38]. Specifically, the competitive binding of 4-HCCA and salicylic acid to Tyrosinase validates CoLD-CoP’s novel application to identifying competitive binders, showcasing the method’s potential to provide insights into enzyme activity and protein-ligand interactions, thereby enhancing the understanding and development of therapeutic agents.

In the context of FBDD, the discovery that tris and imidazole bind cooperatively with GlcNAc is tantamount to identifying synergistically acting fragments. Linkage of these fragments may generate a viable lead compound for further chemical optimization. In a system with a known protein structure, computational and theoretical modeling can guide the assembly of fragments, identified as cooperative binders via CoLD-CoP, into a lead compound. In the absence of a protein structure, differences in chemical shifts between the pair of DOSY spectra input into CoLD-CoP can shed light on how the ligands identified by CoLD-CoP actually bind a target protein.

## 5. Conclusions

The application of CoLD-CoP to identify multiple cooperative and competitive binding interactions with a single pair of experiments, with protein concentrations as low as 4 μM and ligand concentrations as low as 1 mM, facilitates follow-up studies in a FBDD process. This study demonstrates the ability of CoLD-CoP to identify cooperative binders by demonstrating the cooperative binding of GlcNAc and tris to lysozyme. The CoLD-CoP method is particularly suited to FBDD as CoLD-CoP-based analyses can utilize spectra acquired at lower field strengths and on samples with minimal amounts of protein. The methods utilized in this paper exemplify the use of CoLD-CoP in identifying fragments that cooperatively bind a given target to assist in the process of piecing together a promising lead compound based on the results of fragment screening. In general, the application of CoLD-CoP to identify cooperative and competitive binding ligands will facilitate the discovery of compounds modifying protein function, which is critical in both drug development and chemical biology.

## Data Availability

The original data presented in the study, along with scripts used in the analysis presented herein, are openly available in Zenodo at DOI: 10.5281/zenodo.12170717.

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
