# Peer review of "Application of CoLD-CoP to Detecting Competitively and Cooperatively Binding Ligands"

_biomolecules, 2024, doi:10.3390/biom14091136_

Round 1
Reviewer 1 Report
Comments and Suggestions for Authors
The manuscript by Patnala et al. describes the application of CoLD-CoP to detect the cooperative binding of GlcNAc and tris to Lysozyme, and the competitive binding of 4-HCCA and salicylic acid to Tyrosinase.
First, the authors use the previously described CoLD-CoP computational method to predict possible binders. Next, they use diffusion coefficient spectroscopy to determine whether a given small molecule binds to a receptor, as well as to demonstrate and validate the possible competitive or cooperative binding. Finally, modeling tools are used to provide a structural view of the obtained results.
As sed, the use of CoLD-CoP was already described in a previous publication. The inhibitor properties of GlcNAc and Tris to Lysozyme is also known. Thus, this reviewer does not see a sufficient novelty of this work.
The DOSY spectra should be provided and discussed, not only the data analysis and fitting.
Major concerns arise from tables S1-3. Diffusion coefficient is a molecular properties, not atom property. Thus it is not clear how the different signals from different protons of the same molecule have different diffusion coefficient. As an example, from table S1, the proton of the GlcNAc at 1.97 ppm has a Dcoeff. of 5.41, the proton at 3.48 has a Dcoeff. of 6.29; the proton at 3.85 has a Dcoeff of 5.86 etc. etc. The same observation can be done for the free Thiamine and the others molecules.
Those results, the absence of NMR spectra and the lack of novelty are, according to this reviewer, the main weakness of this work.
Author Response
Comments 1: The manuscript by Patnala et al. describes the application of CoLD-CoP to detect the cooperative binding of GlcNAc and tris to Lysozyme, and the competitive binding of 4-HCCA and salicylic acid to Tyrosinase.
First, the authors use the previously described CoLD-CoP computational method to predict possible binders. Next, they use diffusion coefficient spectroscopy to determine whether a given small molecule binds to a receptor, as well as to demonstrate and validate the possible competitive or cooperative binding. Finally, modeling tools are used to provide a structural view of the obtained results.
Response 1: We thank the reviewer for carefully reading our paper and providing us with constructive suggestions that will make our paper clear to scientists with a broad variety of specializations.
Comments 2: As sed, the use of CoLD-CoP was already described in a previous publication. The inhibitor properties of GlcNAc and Tris to Lysozyme is also known. Thus, this reviewer does not see a sufficient novelty of this work.
Response 2: The novel aspects of this work are in (a) demonstrating that CoLD-CoP can be used to detect competitive and cooperative binding rather than just binding and (b) demonstrating that tris and GlcNAc (as well as imidazole and GlcNAc) cooperatively bind lysozyme. To our knowledge the cooperative binding of tris and GlcNAc has not previously been shown. Having a tool capable of identifying such cooperative binding is useful, for example, in fragment-based drug discovery as well as in developing probes of protein chemistry and function.
We have added the following text to indicate that our study validates a novel application of CoLD-CoP and to highlight the utility of this novel application (page 10, paragraph 2, lines 251-255 of the revised manuscript)
Specifically, the competitive binding of 4-HCCA and salicylic acid to Tyrosinase validates CoLD-CoP’s novel application to identifying competitive binders, showcasing the method’s potential to provide insights into enzyme activity and protein-ligand interactions, thereby enhancing the understanding and development of therapeutic agents
We have also added the following text (additions in red) to our Discussion to emphasize that our method can detect multiple competitive or cooperative interactions with a single pair of experiments (page 11, paragraph 1, lines 264-265 of the revised manuscript)
The application of CoLD-CoP to identify multiple cooperative and competitive binding interactions with a single pair of experiments
Comment 3: The DOSY spectra should be provided and discussed, not only the data analysis and fitting.
Response 3: We have added a figure (page 6 of the revised manuscript) providing annotated DOSY spectra for all six mixtures discussed in this paper and briefly discussed them in the figure legend.
Comments 4: Major concerns arise from tables S1-3. Diffusion coefficient is a molecular properties, not atom property. Thus it is not clear how the different signals from different protons of the same molecule have different diffusion coefficient. As an example, from table S1, the proton of the GlcNAc at 1.97 ppm has a Dcoeff. of 5.41, the proton at 3.48 has a Dcoeff. of 6.29; the proton at 3.85 has a Dcoeff of 5.86 etc. etc. The same observation can be done for the free Thiamine and the others molecules.
Response 4: While diffusion coefficient is a molecular property, in DOSY spectra it is not unusual for different signals, arising from different protons in the same molecule, to indicate different diffusion coefficients. While the variation in values that we see is unusually large (e.g. 6.29 is 11% larger than the mean diffusion coefficient we measured for GlcNAc), such variations do occur, especially (in our experience) in the presence of protein. There are multiple reasons for a large variance between diffusion coefficients estimated for the same molecule, including baseline distortions (even after baseline correction) and the non-uniformity of the pulsed field gradients critical to the DOSY pulse sequence. Moreover, diffusion coefficients are inferred from DOSY data using curve fitting techniques that assume the uncertainty in peak height is independent of gradient strength. Failure of this assumption to occur in practice further complicates estimation of the standard error in diffusion coefficients. We have modified the caption for the tables in our supplemental material to include the following explanation
Standard errors in diffusion coefficients are those estimated in least squares fitting of DOSY decay curves for each peak analyzed and typically underestimate the true uncertainty in diffusion coefficients inferred from DOSY data (c.f., M. Delsuc & T. Malliavin, Maximum entropy processing of DOSY NMR spectra, Anal. Chem. 1998, 70, 2146–2148 and J. Guest, P. Kiraly P, M. Nilsson & G.A. Morris GA, Signal-to-noise ratio in diffusion-ordered spectroscopy: how good is good enough? Magn Reson (Gott). 2021, 2, 733-739.).
We also have added the following clarifications (in red, note also the correction of a minor grammatical error in our submitted manuscript) to our Introduction (page 1 last paragraph, continuing onto page 2, second paragraph, lines 39-54 of the revised manuscript). Note that we have also added new references (those listed in the above explanation)
Nuclear Magnetic Resonance (NMR) is a powerful tool with many applications in drug development, including FBDD, pipelines [4,8]. NMR-based techniques, such as Diffusion Ordered Spectroscopy (DOSY) which identifies ligands by their change in diffusion coefficient upon target binding, can identify ligands with KD values ranging from 100 pM to 10 mM [9–11]. However, promising fragment molecules may very well bind with a KD up to 30 mM [12], and experiments to detect fragment binding typically require protein concentrations of one-thousandth or, depending on the type of experiment performed, up to one-tenth of the ligand concentration [13]. Challenges in the use of DOSY to identify ligand binding include the large uncertainty in diffusion coefficients inferred from DOSY data [14,15] as well as the inherent low sensitivity of NMR experiments. Ensuring enough protein is available to form protein-ligand complexes at concentrations sufficient to identify even the weakest-binding promising fragments typically requires around 10 mM ligand concentration and hence 10 μM protein concentration.
This paper demonstrates the application of the Clustering of Ligand Diffusion Coefficient Pairs (CoLD-CoP) method [16], which uses statistical techniques to overcome the large uncertainty in diffusion coefficients inferred from DOSY data, to identify competitively and cooperatively binding ligands.
Reviewer 2 Report
Comments and Suggestions for Authors
The paper titled "Application of CoLD-CoP to Detecting Competitively and Cooperatively Binding Ligands" explores an advanced method for identifying weakly binding ligands in fragment-based drug discovery (FBDD). The authors present the Clustering of Ligand Diffusion Coefficient Pairs (CoLD-CoP) technique, which utilizes Nuclear Magnetic Resonance (NMR) to analyze diffusion spectra and detect competitive and cooperative binding interactions. By comparing ligand diffusion in the presence and absence of a known ligand, the method distinguishes ligands that bind more effectively to a protein either alone or in combination with another ligand. This approach simplifies the process of identifying promising drug fragments, potentially accelerating the development of new therapeutics.
Despite this innovation, the paper requires numerous adjustments to improve the clarity and consistency in the description of the preparation steps and concentrations used. There are some inconsistencies in the description of mixtures and concentrations, which could confuse the readers. Even though the authors depicted this method as a means to use low protein concentration, they do not clarify the amount of protein used for these experiments.
The results are documented but could benefit from a more structured presentation. The discussion on the competitive and cooperative binding findings could be more explicitly linked to the figures and tables for better comprehension.
The explanation from line 73 to 82 is confusing and hard to understand. Please clarify these sentences, possibly with a scheme.
In Figure 1, the filled circles are actually asterisks. Explain the colours legend in the caption.
The relevance of Figure 2, panel A, is unclear. I suggest the authors add a sentence in the text explaining the mixtures used, as it is not clear.
In Figure 4, colouring the tris and the GlcNAc unit differently can help better understand the complex. I suggest renaming Figure 3 to Figure 4 and Figure 4 to Figure 3, to give an overall vision of the complex first, and then go into details. Additionally, you can color the different atoms.
I suggest refining Scheme 2. It is useful for the explanation but needs improvement. The authors can add the real structure of imidazole and GlcNAc in the scheme. There are several Lysozyme-GlcNAc/Lysozyme-Imidazole structures deposited, and the authors can use these as templates.
The discussion does not mention Tyrosinase, making it seem like these experiments were useless. Including a discussion on Tyrosinase would be beneficial.
Author Response
Comments 1: The paper titled "Application of CoLD-CoP to Detecting Competitively and Cooperatively Binding Ligands" explores an advanced method for identifying weakly binding ligands in fragment-based drug discovery (FBDD). The authors present the Clustering of Ligand Diffusion Coefficient Pairs (CoLD-CoP) technique, which utilizes Nuclear Magnetic Resonance (NMR) to analyze diffusion spectra and detect competitive and cooperative binding interactions. By comparing ligand diffusion in the presence and absence of a known ligand, the method distinguishes ligands that bind more effectively to a protein either alone or in combination with another ligand. This approach simplifies the process of identifying promising drug fragments, potentially accelerating the development of new therapeutics.
Response 1: We thank the reviewer for carefully reading our paper and providing us with numerous constructive suggestions.
Comments 2: Despite this innovation, the paper requires numerous adjustments to improve the clarity and consistency in the description of the preparation steps and concentrations used. There are some inconsistencies in the description of mixtures and concentrations, which could confuse the readers. Even though the authors depicted this method as a means to use low protein concentration, they do not clarify the amount of protein used for these experiments.
Response 2: We have revised Scheme #1 and added an additional scheme, as described our response to comment(s) 3. We have also revised the associated text (page 4, paragraphs 1 & 2, lines 103-118 of the revised text) in our methods section to align with the language used in our revised Scheme #1 as well as to include protein concentrations also given in Table 1
Table 1 describes the components of the ligand-only, ligand+protein and ligand+protein+known ligand mixtures used to validate the approach described in Scheme 1. Briefly, solution TL is the ligand mixture used for probing competitive/cooperative binding of mushroom tyrosinase, with TLP being the corresponding ligand+protein sample and TLPK being the corresponding ligand+protein+known ligand sample. TLP was prepared by adding mushroom tyrosinase (to a concentration of 4 μM) to a portion of mixture TL; in the process of preparing mixture TLP, some salicylic acid precipitated resulting in a concentration of only 1 mM in mixtures TLP and TLPK as opposed to the 4 mM concentration in mixture TL. TLPK was prepared by mixing a portion of mixture TLP with 4-HCCA.
Similarly, the ligand-only, ligand+protein and ligand+protein+known ligand samples for probing competitive/cooperative lysozyme binding are labeled LL, LLP and LLPK. LLP was prepared by adding lysozyme (to a concentration of 1 mM) to a portion of mixture LL, and LLPK was prepared by adding GlcNAc to mixture LLP. The solvent for all solutions was D2O. Solutions LL, LLP and LLPK were adjusted to pH = 6 using NaOH. The pH of solutions TL, TLP and TLPK was not adjusted.
Comments 3: The results are documented but could benefit from a more structured presentation. The discussion on the competitive and cooperative binding findings could be more explicitly linked to the figures and tables for better comprehension.
The explanation from line 73 to 82 is confusing and hard to understand. Please clarify these sentences, possibly with a scheme.
Response 3: We have extensively revised (revisions shown in red) the entire first paragraph of our Materials and Methods section (page 2, lines 69-86 of the revised manuscript), breaking it up in the process
Scheme 1 outlines the steps of our approach. Briefly, our approach involves making three different mixtures: one containing only the putative ligands (the ligand-only sample: Step 1.A), one with putative ligands and a protein of interest (the ligand+protein sample: Step I.B), and one with the putative ligands, a protein of interest, and a known ligand binding the protein of interest (the ligand+protein+known ligand sample: Step I.C). Following the acquisition and processing of DOSY data for the ligand-only, ligand+protein and ligand+protein+known ligand samples (Step 2), the DOSY peak lists are compared using CoLD-CoP [16].
As described in Scheme 1, substituting the ligand+protein+known ligand peak list for the ligand-only peak list typically used in running CoLD-CoP (c.f. Step 3.B.i) identifies ligands diffusing more slowly in the absence of the known ligand (Step 3.B.ii), whereas substituting the ligand+protein peak list for the ligand-only peak list and the ligand+protein+known ligand peak list for the ligand+protein peak list identifies ligands diffusing more slowly in the presence of the known ligand (Step 3.B.iii). As illustrated in Scheme 2, ligands that diffuse more slowly in the absence of the known ligand bind the protein of interest more effectively without the known ligand and hence compete with it for protein binding. Conversely, ligands that diffuse more slowly in the presence of a known ligand bind the protein of interest better with the known ligand, indicating cooperative binding.
Note that the above text refers to a revised Scheme 1 that replaces bullet points with lettering and numbering to ensure a clearer correspondence between Scheme 1 and the text. As suggested by the reviewer, we have included an additional schematic (Scheme 2) to illustrate the (now revised) explanation from line 73 to 82 in our original submission.
Comment 4: In Figure 1, the filled circles are actually asterisks. Explain the colours legend in the caption.
Response 4: We felt that at the resolution readers would typically view the figures, the asterisks would appear as filled circles. Since the reviewer is able to see the asterisks (which is how the MATLAB toolbox implementing CoLD-CoP displays them), we have modified our figure legend (page 7, lines 174-189 of the revised manuscript), which now also includes a description of the colors used (note this is now Figure 2, due to a figure added based on another reviewer’s suggestion)
Figure 2. CoLD-CoP comparisons: circles and asterisks are colored (automatically by the MATLAB toolbox implementing CoLD-CoP) based on the cluster to which CoLD-CoP assigns them. Ideally, each cluster represents one and only one molecule, but molecules of similar size whose diffusion coefficient does not change between the solutions being compared may not be distinguishable based on paired diffusion coefficients. Open circles indicate diffusion coefficients that do not significantly change between the compared solutions while (filled) asterisks indicate a significant result. (a) CoLD-CoP comparison of solutions TL and TLP. In this comparison, the chemical shifts in the cluster denoted by blue asterisks are assignable to salicylate; that they are asterisks (rather than open circles) indicates that binds tyrosinase (b) comparison of solutions TLP and TLPK to detect competitive binding as described in Scheme 1, (c) Same as (b), but zoomed in to show competitive binding more clearly: asterisks indicate clusters with significantly higher diffusion coefficients in the presence of 4-HCCA, indicating competitive binding. Each significant cluster (yellow and red asterisks) is associated with chemical shifts assigned to salicylate. Note that salicylate, tartrate, and tris all have similar molecular masses and hence similar diffusion coefficients. This approach cannot distinguish between tris and tartrate (empty cyan circles), but does distinguish (albeit dividing it into two clusters) salicylate as competing with 4-HCCA for tyrosinase binding.
Comment 5: The relevance of Figure 2, panel A, is unclear. I suggest the authors add a sentence in the text explaining the mixtures used, as it is not clear.
Response 5: We have revised the figure legend (now for Figure 3, page 8, lines 198-206) for panel A to reference our revised Scheme 1 to make it more clear the purpose of this comparison.
(a) The purpose of this comparison, performed as described in Scheme 1, Step 3.B.ii, is to detect any ligands binding competitively with GlcNAc via CoLD-CoP analysis of solution LLP (Lysozyme + Ligands: Caffeine, Citrate, Imidazole, Tartrate, Tris, Tryptamine) against LLPK (Lysozyme + Lig-ands + Known Ligand: GlcNAc). CoLD-CoP does identify any ligand as competing with GlcNAc for lysozyme binding. (b) A similar comparison performed as described in Scheme 1, Step 3.B.iii.
Comments 6: In Figure 4, colouring the tris and the GlcNAc unit differently can help better understand the complex. I suggest renaming Figure 3 to Figure 4 and Figure 4 to Figure 3, to give an overall vision of the complex first, and then go into details. Additionally, you can color the different atoms.
Response 6: We have swapped these figures and also colored the tris and GlcNAc green and blue, respectively. We were unable to produce a clean looking figure with different colors for the atoms, however.
Comments 7: I suggest refining Scheme 2. It is useful for the explanation but needs improvement. The authors can add the real structure of imidazole and GlcNAc in the scheme. There are several Lysozyme-GlcNAc/Lysozyme-Imidazole structures deposited, and the authors can use these as templates.
Response 7: We thank the reviewers for finding this scheme useful. We were unable to find appropriate structures for either the lysozyme-GlcNAc or lysozyme-imidazole complexes. However, unrestricted docking docks imidazole to the known lysozyme dimerization interface. Additionally, we already have the structure of GlcNAc docked to lysozyme. We thus were able to refine scheme 3 (renumbered due to another addition suggested by this reviewer) as the reviewer suggests. However, we still needed to keep the hexagons and pentagons, because the ligand structures were not obvious enough without being highlighted. The caption for this scheme (page 10, lines 240-245 of the revised manuscript) now reads (additional text in red)
Scheme 3. Model for cooperative GlcNAc and imidazole binding. (a) Lysozyme monomer (active), (b) Lysozyme dimer (inactive), (c) Lysozyme monomer + imidazole (competitively inhibited), (d) Lysozyme monomer + GlcNAc (competitively inhibited), (e) Lysozyme monomer + 2 imidazole, (f) Lysozyme monomer + GlcNAc + imidazole. GlcNAc and the imidazole at the dimer interface are shown where they dock to lysozyme. GlcNAc and imidazole structures are highlighted by purple hexagons and yellow pentagons, respectively, for clarity.
We also modified the Materials and Methods section to mention our docking of imidazole (additional text in red, page 5, sole paragraph, lines 128-130)
iGEMDOCK [30] docked tris, imidazole and GlcNAc to lysozyme (PDB ID 2D4K15). The ternary complex (tris-GlcNAc-lysozyme) was manually adjusted using the Schrödinger [31] interface.
Comments 8: The discussion does not mention Tyrosinase, making it seem like these experiments were useless. Including a discussion on Tyrosinase would be beneficial.
Response 8: The primary role of the tyrosinase experiments was to demonstrate the novel application of CoLD-CoP to identifying competitive (and cooperative) binding on a system with two ligands known to bind a particular protein competitively. We have modified the Discussion (page 10, paragraph 2, lines 247-255) to include text in red (note the changed reference number due to adding two references to our paper as part of our response to another reviewer’s comment):
A putative ligand identified as binding competitively with a lead fragment (in this example, 4-HCCA and GlcNAc) would be a poor choice to assemble with the lead fragment in the context of FBDD. However, the identification of competitive binders, which may themselves be agonists, to endogenous ligands/substrates provides protein chemists with probes of protein structure and function [38]. Specifically, the competitive binding of 4-HCCA and salicylic acid to Tyrosinase validates CoLD-CoP’s novel application to identifying competitive binders, showcasing the method’s potential to provide insights into enzyme activity and protein-ligand interactions, thereby enhancing the understanding and development of therapeutic agents.
Round 2
Reviewer 2 Report
Comments and Suggestions for Authors
Please check Scheme 3 because it is not in the main text
Author Response
Reviewer Comment #1: Please check Scheme 3 because it is not in the main text
Response #1: Thank you for noticing this omission. We had included this scheme in our revised manuscript, but, due to some sort of technical glitch, it was not included in the document generated by the document submission system. We have reinserted our image for Scheme 3 on line 249, right above the scheme legend.